# Determination of Pesticide Residues in IV Range Artichoke (*Cynara cardunculus* L.) and Its Industrial Wastes

**DOI:** 10.3390/foods12091807

**Published:** 2023-04-26

**Authors:** Francesco Corrias, Nicola Arru, Alessandro Atzei, Massimo Milia, Efisio Scano, Alberto Angioni

**Affiliations:** 1Department of Life and Environmental Science, Food Toxicology Unit, University of Cagliari, University Campus of Monserrato, SS 554, 09042 Cagliari, Italy; nicola.arru.logica@gmail.com (N.A.); alessandro.atzei@unica.it (A.A.); max_milia@hotmail.it (M.M.); aangioni@unica.it (A.A.); 2Faculty of Agraria, University of Sassari, 07100 Sassari, Italy; efisiscano@gmail.com

**Keywords:** artichoke, pesticide residues, industrial processing, waste, LC-MS/MS

## Abstract

Fourth-range products are those types of fresh fruit and vegetables that are ready for raw consumption or after cooking, and belong to organic or integrated cultivations. These products are subject to mild post-harvesting processing procedures (selection, sorting, husking, cutting, and washing), and are afterwards packaged in packets or closed food plates, with an average shelf life of 5–10 days. Artichokes are stripped of the leaves, stems and outer bracts, and the remaining heads are washed with acidifying solutions. The A LC-MS/MS analytical method was developed and validated following SANTE guidelines for the detection of 220 pesticides. This work evaluated the distribution of pesticide residues among the fraction of artichokes obtained during the industrial processing, and the residues of their wastes left on the field were also investigated. The results showed quantifiable residues of one herbicide (pendimethalin) and four fungicides (azoxystrobin, propyzamide, tebuconazole, and pyraclostrobin). Pendimethalin was found in all samples, with the higher values in leaves 0.046 ± 8.2 mg/kg and in field waste 0.30 ± 6.7 mg/kg. Azoxystrobin was the most concentrated in the outer bracts (0.18 ± 2.9 mg/kg). The outer bracts showed the highest number of residues. The industrial waste showed a significant decrease in the number of residues and their concentration.

## 1. Introduction

*Cynara cardunculus* L. is an herbaceous perennial plant belonging to the family of Asteraceae and widespread in all Mediterranean countries. It is cultivated as a poly-annual crop in different soils and climate conditions [1]. Europe is the leading producer with 42.5%, followed by Africa (35%). Italy represents the primary world producer of artichokes (372 ktons/year), and other leading countries are represented by Egypt (313 ktons/year), Spain (206 ktons/year), and Algeria (124 ktons/year) for a total worldwide production of about 1470 ktons in 2021 [2]. The edible part is represented by the inflorescence forming on the top of the central and lateral stems, and the hearth is surrounded by involucral bracts imbricated as rose petals. Artichoke is rich in bioactive compounds, polyphenols, inulin, vitamins, minerals, and fiber; in contrast, it has a low content of fats [3,4]. Therefore, it is considered a health-promoting food, eaten worldwide raw, cooked, or canned [5]. During the preparation of IV-range products, artichokes generate a considerable volume of waste (80–85% of total harvested plant biomass), represented by the not edible external bracts, stems, and leaves. However, this waste represents only around 15% of the whole plant left on the field at the end of the harvesting season [6,7]. Fourth-range products are fresh fruit and vegetables ready for consumption, with an average shelf life of between 5 and 10 days. Before packing, these products undergo minimal processing steps (barely treated), such as selection, sorting, husking, cutting, and washing. Regarding artichokes, the most common practices are eliminating the stem with the leaves and the outer bracts, and washing them with acidifying solutions to avoid external oxidation [8]. Artichoke cultivation faces different disease threats, which could influence field yields. The most harmful pathogen is the fungus *Verticillium dahliae* Kleb, which leads to wilting and leaf fall [9,10,11]; other pathogens are represented by *Leivellula taurica* (powdery mildew), *Sclerotinia sclerotiorum* (white mold), and *Botrytis cinerea* (gray mold); moreover, sucking insects, such as aphids, mites, and thrips, can infest artichokes. Pesticides are widely used in open fields to overcome these pathologies and minimize crop losses, leading to possible multiresidue contamination, even if applied in Good Agriculture Practices (GAP) [12,13]. The processing of many foods can influence the concentration of pesticide residues in food products [14,15,16,17,18,19,20]. Graziela et al. and Cengiz et al. evaluated the ability of washing and peeling, hydrogen peroxide, and ozone application in reducing the concentration of pesticides in the tomato, respectively [17,21]. Corrias et al. studied the influence of an entire industrial process on the transfer of pesticide residues from raw tomatoes on the processing products (purée, triple concentrated paste, fine pulp, and diced tomatoes) [22]. Alister et al. assessed the pesticide residue processing factor (PF) from plums to prunes [14], whereas Bonnechere et al. studied the PF for boscalid, deltamethrin, mancozeb, iprodione, and propamocarb on spinach after several household and industrial processes [23]. Moreover, Corrias et al. established the effectiveness of the technological process of winemaking to decrease pesticide residues compared to the raw material [24]. Multiresidue pesticide analysis in the edible part of artichokes has been investigated by LC-MS and GC-MS methods [25,26,27]. Viana et al. optimized a matrix solid-phase dispersion method for the analysis of pesticide residues in artichokes and other vegetables [25]. Machado et al. performed the determination of 98 pesticides in artichoke leaves and fruits, both by LC-MS and GC-MS [26]. No articles dealing with the multiresidue analysis of pesticides in the different portions of the artichoke plant or the influence of industrial processing on their concentration are present in the literature. In this paper, we developed and validated a multiresidue LC-MS/MS coupled with a modified QuEChERS extraction method for determining 220 pesticides on artichokes after IV range processing (Appendix A). Residues of pesticides authorized and not for the use on artichoke were analyzed in samples of whole fresh artichoke, in the edible part (head), and in the different portions of waste produced (leaves, stems, and bracts); moreover, samples left in the field at the end of the harvest were analyzed for pesticide pollution for a possible reuse of the waste in the food, cosmetic, or pharmaceutical fields.

## 2. Materials and Methods

### 2.1. Chemicals and Reagents

Acetonitrile (ACN) and methanol (MeOH) were LC/MS grade solvents purchased from Sigma Aldrich (Milan, Italy). Ammonium formate 5 M (Part number: G1946-85021) and formic acid (reagent grade > 95%) were from Agilent technologies and Honeywell (Sigma Aldrich), respectively. Certified analytical standards (≥98.0% purity) of 220 pesticides at 100 mg L^−1^ in ACN (LC/MS Pesticide Comprehensive Test Mix Kit–Part number: 5190-0551) were purchased from Agilent Technologies (Appendix A). The intermediate solution of pesticide mix was prepared at 1 mg L^−1^ in ACN. The five-point matrix-matched calibration curves were prepared daily by serial dilution of the 1 mg/L^−1^ intermediate solution in blank artichoke extract (10 mL). A MilliQ Millipore purification system was used to produce water with a conductivity less than 18.2 MΩ, (MilliQ integral, Merck, Milan, Italy).

### 2.2. Samples Collection and Processing

One hundred and fifty fresh artichokes (cv. Tema 2000) were randomly collected in a field of 1 ha located in Samassi (Sardinia, Italy) in March 2022. Artichokes were checked for any damage or organoleptic alterations. Fifty artichokes were taken to the laboratory and separated into outer bracts, stems, heads, and leaves. Each subsample was chopped using a stainless-steel food cutter mixer (K55, Electrolux Professional, Pordenone, Italy) and subjected to the analytical procedure. The other artichokes were taken to the factory and subjected to industrial processing. Stems and leaves were removed manually, whereas the outer bracts were removed from the head with a turning machine. The obtained waste was chopped using a Viking GE–105 electric shredder (Viking GmbH, Langkampfen, Austria). The homogenized waste and peeled heads were analyzed in the laboratory. At the end of the harvesting season (May 2022), twenty fresh plants were collected from the same field, separated from the roots, chopped, homogenized, and taken to the laboratory for analysis. Specialized technicians performed field treatments following IPM strategies for artichokes. Samples belonging to selected organic fields were collected for blank control matrix extracts.

### 2.3. Sample Preparation

Individual samples (10 g) were homogenized and weighed in a 50 mL test tube plus 10 mL of ACN. After being vigorously shaken in a vortex (Reax Top, Heidolph, Schwabach, Germany) for 1 min, the first QuEChERS salts were added to the test tubes (6.5 g, Part No: 5982-6650) The samples were agitated in a vortex for 2 min and in a rotatory shaker for 15 min, and centrifuged at 3154× *g* and 10 °C for 15 min (Centrifuge 5810 R, Eppendorf AG 22331, Hamburg, Germany). Five milliliters of the extracting solvents were transferred to a 15 mL test tube together with the second QuEChERS salts (1 g, Part No: 5982-5056, Agilent, Milan, Italy). The test tubes were subjected to the previous mixing procedure, and so the organic solution was filtered at 0.45 µm (PTFE, Thermo Scientific, Roma, Italy) and transferred in a vial for LC-MS/MS analysis.

### 2.4. UHPLC-MS/MS Analysis

A UHPLC Agilent 1290 Infinity II LC coupled with an Agilent 6470 Triple Quad LC-MS/MS mass detector with a MassHunter ChemStation was used. The instrument’s analytical conditions were in accordance with Corrias et al. [26], with only small modifications related to the increased number of pesticides analyzed (Appendix A). Briefly, a binary gradient composed of a 5 mM ammonium formate + 0.1% formic acid aqueous solution (A) and a 5 mM ammonium formate + 0.1% formic acid methanolic solution (B) was set as follows: t = 0 A 95%, t = 1 A 95%, t = 3.00 min A 55%, t = 16 min A 5%, t = 22.50 min A 5%, t = 22.60 min A 95%, with a post-run of 6 min (95% A), and with the total duration of the run being 28.60 min. One μL of sample volume was injected in positive mode. Mass detector gas and sheath-gas temperature were 120 °C and 325 °C, whereas sheath-gas flow, nebulizer, and positive capillary were set at 12 L min^−1^, 45 psi, and 3500 V, respectively. Data were acquired in Dynamic MRM.

### 2.5. Method Validation

The analytical method validation was carried out following the SANTE Guidelines [28]. The MRM chromatogram of the eluting mixture, the control matrix, and spiked matrices at the LOQ (limit of quantification) level were used for selectivity evaluation. The absence of instrumental response at the retention times of the selected analytes was selected as a confirmation criterion of the method’s selectivity. The instrumental LOD (limit of detection) and LOQ were calculated according to SANTE guidelines. Linearity was assessed using five-point calibration curves performed in solvent and blank control matrix extracts. A coefficient of determination (r^2^) above 0.990 was considered adequate. The ME represented by possible interfering signals arising from the sample was evaluated using the following formula: ME (%) = (A × 100/B) − 100
where ME represents the matrix effect, A the slope of the matrix-matched calibration curve, and B the slope of the solvent calibration curve in ACN.

Repeatability (RSDr) and reproducibility (RSDwR) were evaluated by analyzing six blank control matrix samples spiked with the mixed multiresidue standard in one day (RSDr) and three separate days (RSDwR). Recovery assays were performed by spiking the blank control matrix with the mixed multiresidue pesticide standard solution (three replicates each) at the concentration reported in Appendix A.

### 2.6. Processing Factor

The processing factor (PF) was calculated diving the residue level in the IV gamma processed commodity (industrial waste) with those found in the raw commodities (outer bracts) [29]. For the residue value < LOQ, no PF was calculated. PF > 1 indicates an increase in the residue during processing, but, on the contrary, a PF < 1 indicates a decrease in the residue.

## 3. Results and Discussion

### 3.1. Method Validation

The LC-MS/MS method allowed the simultaneous determination of 220 pesticides on artichoke samples (Figure 1). No interfering peaks were found in the time interval of interest for the analytes. The matrix effect (ME) showed that 7% of the pesticide’s instrumental response was influenced by the coextracted compounds of the artichoke matrix: among these compounds, 46% were suppressed, and 54% increased (Appendix A). Therefore, five-point calibration curves were prepared in blank artichoke matrix extracts ranging from 0.005–0.50 mg/kg.

The correlation coefficients (r^2^) ranged from 0.990 to 1.000 (Appendix A), and almost 91% of the LOQs were below 0.005 mg/kg. Avermectin B1a, dimethoate, ethoprophos, and etofenprox had LOQs higher than the MRL set, but they are not authorized in Italy for use on artichokes. Apparent recoveries were performed at 0.005 mg/kg, 0.025 mg/kg, and 0.25 mg/kg (Appendix A). Almost all pesticides showed good recoveries according to SANTE indications. However, some compounds were far below the attended recoveries at 0.005 mg/kg, and aldicarb, azamethiphos, carbendazim, dimethoate, dioxacarb, etofenprox, and fenuron showed extremely low recoveries below 36%, whereas azinphos-ethyl, butocarboxim, ethidimuron, flubendiammide, nitenpyram, and pirimicarb showed values around 65%. Chloridazon, cymiazole, ethoprophos, metribuzin, thiofanox, and vamidothion recoveries ranged 51.3% ± 10.7 (% ± RSD). In contrast, avermectin B1a showed at 0.005 mg/kg recoveries values above SANTE recommendations (% ± RSD), (Appendix A). Repeatability and reproducibility were carried out at the LOQ values of each pesticide and showed in all trial values below 20% (Appendix A). Regarding pesticides authorized on artichokes, all 22 compounds (6 insecticides, 4 herbicides, and 12 fungicides) showed LOQ levels well below the MRL (Appendix A). Acetamiprid was the only analyte showing recovery at 0.005 mg/kg not complying with SANTE indications (27.1%), whereas the other pesticides showed recovery values at LOQ level ranging from 74.3 ± 8.5% (cymoxanil) to 117.9 ± 9.6% (dimethomorph) (Appendix A). The limit of quantification for acetamiprid was set at 0.025 mg/kg (recovery 94.0 ± 3.5%), 28 times lower than its MRL (0.7 mg/kg) (Appendix A). Difenoconazole, with an increase in the signal of about 350%, was the only pesticide affected by the matrix effect (Appendix A). Only a small number of articles can be found in literature dealing with pesticide residues on artichokes. Machado et al. (2017) compared different extraction methods on 15 pesticides selecting as the best option a modified QuEChERS method. The method has been validated on 85 pesticides, 35 in GC-MS, and 63 in LCMSMS; the recoveries fall under SANTE recommendations; however, they were primarily situated in the lower range of recoveries. This study did not consider the compounds with low recoveries found in our paper [26]. Almela et al. (2020) analysed eleven pesticides in GC-MS/MS and LC-MS/MS after application in field trials. The residues were determined only on artichoke heads at commercial size. Only diphenconazole showed residues above the quantification limit, and the other pesticides were not revealed [12]. Cabras et al. (1996) analysed the behaviour of three pesticides by GC-MS on two different shape cultivars, Spinoso Sardo and Masedu [27], whereas Hassen et al. (2018) analysed dimethoate and pentachlorophenol (PCP) in relation to pesticide contaminated artichoke soils [30]. The recoveries obtained in artichokes were compared with those obtained in tomatoes and vernaccia wine, as reported in previous papers [22,24]. A total of 84 pesticides were in common in all matrices. However, only in artichokes could low recoveries be detected, denoting a difficulty in the extraction of the active principles when performed at the lowest spiking concentration. This fact could be attributed to the different composition of the artichoke’s matrix. After homogenization, tomato and its by-products and vernaccia are almost completely liquid, providing a large exchange surface to the extraction solvent, but artichokes have a high percentage of fiber (around 6%) that can hinder the extraction or adsorb pesticide residues [31].

### 3.2. Analysis of Fresh and Processed Artichoke and Artichoke by-Products

Among the 220 pesticides studied, only five compounds were detected and quantified (Table 1). Pesticide residues were represented by one herbicide (pendimethalin) and four fungicides (azoxystrobin, propyzamide, tebuconazole, and pyraclostrobin). Pendimethalin residues were found in all artichoke samples. Analysis showed higher levels in the plant (0.30 ± 6.7, mg/kg ± RSD%) followed by roots and leaves, whereas stems and outer bracts showed only detectable levels, and no residues were found in the peeled heads. Among fungicides, azoxystrobin showed higher levels on outer bracts at 0.18 ± 2.9 mg/kg ± RSD% and in the industrial waste (bracts + leaves + stems) at 0.029 ± 15.2 mg/kg ± RSD% (Table 1). Tebuconazole showed quantifiable residues in the roots (0.004 ± 6.7 mg/kg ± RSD%) and levels below the LOQ in the outer bracts. In contrast, propyzamide and pyraclostrobin showed no quantifiable levels in the outer bracts. The outer bracts from IV range processing resulted in the most polluted artichoke portion with five residues of different pesticides, followed by the roots and plants from the field. The heads never showed pesticide residues (Table 1). A cone-shaped head characterizes Tema CV artichokes, with the bracts closely adherent to each other; this morphology protects the inner bracts and the hearts from contamination from external pollutants. These data agree with previous studies that evaluated the distribution of three pesticides in the outer bracts and flower heads of two artichoke cultivars with different head shapes (Spinoso Sardo vs. Masedu) after spray treatment. The authors showed that head shape dramatically affects the residue amount on artichokes, and the cone shape of Spinoso Sardo had lower values of pesticide residues than the calyx shape of Masedu [27]. Artichoke industrial waste is composed of more than 95% of the outer bracts and the remaining part of the stems (~2%) and leaves (~3%); considering this fact, we should have found significant levels of residues in the industrial waste after IV range industrial processing (Table 1). On the contrary, the analysis showed levels of pendimethalin below the LOQ, and the amount of azoxystrobin was almost 85% lower (Table 1). This result was confirmed by the calculation of the processing factor for azoxystrobin residues quantified in the outer bracts vs. the waste from industrial processing. The PF accounted for 0.16, indicating a decrease in the residues during the IV gamma processing (Table 1). Different studies showed that the comminution processes, such as chopping, shredding, and crushing, can release enzymes and acids, which may increase the degradation of some pesticides [19]. Moreover, the waste material is highly contaminated by microorganisms, and fermentation processes can take place degrading pesticide residues [32]. In addition, the heat released during the shredding process can cause the degradation of heat-sensitive pesticides [20]. During the industrial processes, high amounts of raw matrices belonging to different fields with different pesticide treatments are merged in huge bins. This fact can decrease pesticide residues related to a dilution effect, as already reported during the processing of raw tomatoes into final products (purée, triple concentrated paste, fine pulp, and diced tomatoes) [22] and of grapes into wine [24,33]. During artichoke IV-grade industrial processing, hundreds of artichokes from different fields are processed simultaneously, and their waste is mixed in the collecting tank before further activities. The merging of various parts from diverse areas with varying pollution levels could lead to the decrease in pesticide residues in artichoke waste found in our samples, which is related to the dilution effect. At the end of the harvest, stalks, leaves, and roots remain in the field. Many studies have highlighted the good prospect of feed enriched with artichoke silage from field waste. Dairy cattle showed good performance with increased total milk production and content in macronutrients [34]. On the other hand, lactating ewes produced milk with high organoleptic values but less suitable for cheese production [35]. The high fiber content of these wastes has limited their use in ruminants, which have the ability to degrade it [36]. Moreover, the aliquot of the artichoke silage added could not overcome the 30% of the feed mass. Therefore, only a minor part of the waste left on the field is used as feed for sheep and cows after sun drying, whereas the main part represents a real waste and is usually burned. Waste recovery in a circular economy perspective requires assessing not only its biochemical composition for a nutritional evaluation in terms of overall value, but also to acquire the toxicological characteristics of the waste for human safety assessment. The evaluation of the contamination from pesticide residues represents a fundamental step. The artichoke analyzed from the field has been subjected to treatments with pendimethalin and azoxystrobin. The herbicide pendimethalin showed residues in the remaining plant (stalks and leaves) at a concentration six times higher than the MRL for the edible part (0.05 mg/kg). Azoxystrobin showed higher values in the roots, whereas pesticide residues were negligible for tebuconazole (Table 1). Pendimethalin can be used in pre-transplant, pre-regrowth, or on growing crops (only in the inter-rows and around the stumps). This herbicide is persistent in the ground for 3–6 months, and the roots and shoots can absorb it. Root contamination can occur during field applications or by absorption of persistent pollutants left on the ground from previous treatments, depending on the root concentration factor and its translocation factor in the plant [37]. However, once absorbed into plant tissues, translocation of pendimethalin from root to shoot is limited [38]. Therefore, a post-emergence application on mature crops could explain the high concentration of pendimethalin found in the leaves. Artichoke plants exhibit large basal leaves (up to 1 m in length), and previous studies on high leaf contamination related to leaf shape were reported for other commodities and pesticides [39,40]. Azoxystrobin is a fungicide with systemic and curative properties used on artichoke against powdery mildew and downy mildew. In Sardinia the environmental conditions with high humidity and temperate weather facilitates the spread of these diseases by requiring the use of fungicides. Data obtained on stalk and leaves and on the roots were in accordance with what was reported from previous authors on other crops. Ju et al. (2019) investigated the uptake mechanism in roots under laboratory conditions and the translocation rate of azoxystrobin to the aerial portions in wheat plants [41]. Azoxystrobin primarily accumulated in roots, but its migration rate from roots to stems was limited. Fate, behavior, and metabolization of pesticides after open field application and the consequent spread to the environment are a major aspect of pesticide risk assessment [42]. Thus, monitoring studies on authorized and unauthorized pesticides and their metabolites in different crops are strongly recommended. In this work, all the compounds found are authorized for the use on artichokes in Italy, suggesting a proper use of the plant protection products and a good adherence by operators to the Integrated Pest Management strategies. However, the other pesticides detected (propyzamide, tebuconazole, and pyraclostrobin) were not applied during the current producing season, and the presence of residues was therefore related to treatments on artichoke or on other crops cultivated in the same field in the previous year. In addition, contamination could not be attributed to spray drift, since the fields were isolated from other cultivations. No other studies dealing with the analysis of pesticide residues on the different portions of artichoke or on its waste after industrial processing or from the field were found in the literature.

## 4. Conclusions

This work has developed and validated a multi-residue method for simultaneously determining 220 pesticides on artichoke samples; among these, 22 compounds are authorized on artichokes in Italy. The procedure was specific and robust, showing good recovery data and limits of quantification far below the MRL set for about the 91% of the selected pesticides. Substances contained naturally in artichoke influenced the instrumental response in 7% of the analytes (46% suppressed and 54% enhanced). Fresh artichokes (heads, leaves, stems, and external bracts), industrial production waste, and waste deriving from the field (stalks + leaves, and roots) were subjected to analysis. Five residues (pendimethalin, azoxystrobin, propyzamide, tebuconazole, and pyraclostrobin) were found in the different artichoke portions and waste. Although all these pesticides are authorized on artichokes, only pendimethalin and azoxystrobin were applied during field treatments. Thus, even if treatments were conducted in good agricultural practice (GAP), some residues could remain in the by-products, and also belong to previous treatments. Due to their peculiar shape, the heads were the less polluted part of the artichoke, whereas the outer bracts, exposed to the external environment, resulted in the most polluted. Pendimethalin and azoxystrobin were the most frequent and most abundant residues. The industrial process (shredding and mostly the dilution effect) significantly decreased the number of residues and their concentration in post-processing and waste. The aerial part of the residual artichoke plant from the field showed the most concentrated residue among all the samples. Artichoke waste is a product with a high content of bioactive substances available to obtain phyto-complexes for nutraceutical and pharmaceutical purposes. Applying the circular economy principle to artichoke cultivation, the waste could be transformed into a resource with a high added value; however, waste pollution from persistent pesticides should be carefully controlled to avoid the transfer of toxic pesticides in the nutraceutical extracts.

## Figures and Tables

**Figure 1 foods-12-01807-f001:**
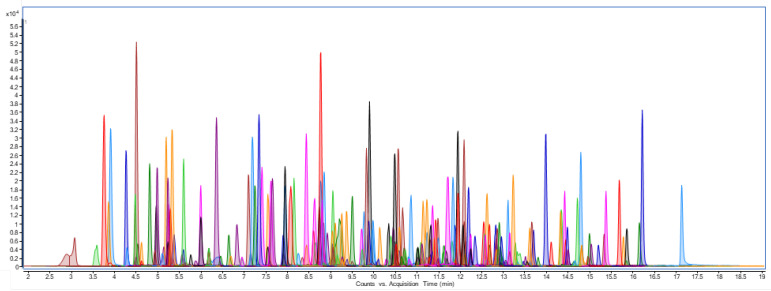
MRM chromatogram of artichoke matrix fortified at LOQ with 220 pesticides.

**Table 1 foods-12-01807-t001:** Pesticide residues concentration (mg/kg ± RSD%) in fresh artichokes and waste from industrial processing and the field.

		Fresh Artichoke				Industrial Processing		Field Waste		
Compound	MRL	Stems	Outer Bracts	Leaves	Heads	Heads	Waste	Stalks and Leaves	Roots	PF *
Pendimethalin	0.05	0.004 ± 15.8	0.005 ± 8.8	0.046 ± 8.2	<LOQ	<LOQ	<LOQ	0.30 ± 6.7	0.038 ± 28.4	
Azoxystrobin	5.00	-	0.18 ± 2.9	-	-	-	0.029 ± 15.2	0.007 ± 3.9	0.027 ± 11.0	0.16
Propyzamid	0.02	-	<LOQ	-	-	-	-	-	-	
Tebuconazole	0.6	-	<LOQ	-	-	-	-	-	0.004 ± 6.7	
Pyraclostrobin	3	-	<LOQ	-	-	-	-	-	-	

* Processing factor.

## Data Availability

The datasets generated for this study are available on request to the corresponding author.

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
