# Peer review of "Determination of Pesticide Residues in IV Range Artichoke (Cynara cardunculus L.) and Its Industrial Wastes"

_foods, 2023, doi:10.3390/foods12091807_

Round 1
Author Response
Reviewer 1
The manuscript “Determination of pesticide residues in IV range artichoke (Cynara cardunculus L) and its industrial wastes” have the necessary quality and the data are relevant. I would like to point out the following suggestions and corrections:
Abstract
1)Revise the abstract, providing more relevant information about the results obtained and the novelty of this study.
Answer: The text was completely revised.
2) Line 21 - 24: I suggest replacing the word polluted with contaminated (line 63 and others too).
Answer: The text was completely revised, and the sentence was changed.
Introduction
3) Replace Kton with kton in the text.
Answer: the text was corrected.
4) It would be interesting to insert in the introduction that the pesticides recommended for artichoke cultivation and the respective maximum residue limits allowed are presented in table S1.
Answer: The information was included in the introduction.
5) Please put these pesticides in bold in addition to the asterisk. It makes it easier for the reader to identify them.
Answer: The pesticides with the asterisk were put in bold.
6) Specify below this table what the letters I, E, F, etc. mean in the "Type" column.
Answer: a note with the information was added.
Materials and Methods
7) Line 68: Suggestion: Replace M or mM with mol L-1 or mmol L-1 in the text.
Answer: Sorry, we did not follow the suggestion.
8) Line 112 and 113: Replace “T” with “t”– (Usually “T” is for temperature). Check information: T = 1 to 95%.
Answer: The text was corrected, and T replaced with t.
9) Line 116: Replace 120°C and 325°C with 120 °C and 325 °C.
Answer: The text was corrected.
10) Line 117: Replace 12 L min-1, with 12 L min-1.
Answer: The text was corrected with 12 L min-1.
11) Line 126: “The instrumental LOD and LOQ were three and ten times the signal to-noise ratio (S/N).”
Answer: the text was change in agreement with the Editor and the reviewer.
12) Question: Were the linearity and LOD and LOQ of the method also tested? Or were they evaluated only for the instrument?
Answer: the values reported, represent the LOD and LOQ of the method and not the instrumental LOD and LOQ.
13) Line 129: Suggestion: Replace "correct" with "adequate".
Answer: The text was corrected, and the word correct replace with adequate.
Results and discussion
Lines 140 - 155: 3.1. Method validation
14) It would be interesting to make a quick evaluation about these results regarding the pesticides recommended for artichoke cultivation.
Answer: We agree with the reviewer, and a specific evaluation of the method validation of the pesticides authorized on artichokes along the discussion was added.
Line 165: Analysis of fresh and processed artichoke and artichoke by-products
It should be noted that the results obtained bring a relevant contribution to the environmental area, in terms of contamination by pesticides.
Answer: We agree with the reviewer.
15) It would also be interesting to point out that the pesticides found (Table 1) are authorized for this crop. This indicates that there was no misuse of other pesticides, nor contamination from pesticide applications on other crops, etc.
Answer: We agree with the reviewer and this topic was added in the discussion. However, propyzamid, tebuconazole, and pyraclostrobin were not applied during the current producing season, therefore the presence of residues was related to treatments on artichoke or on other crops cultivated in the same field in the previous year.
Line 224 – 4. Conclusions
Line 225: “This work has developed and validated…”
Question: Was the method developed and validated or only validated in this work (line 58)?
Answer: The proposed method was developed and validated.
Reviewer 2 Report
It is very interesting to study the effect of pesticide residue in artichoke during industrial processing with queQuEChERS method. It would be better if use field sample with pesticide spraying, not just could determine 220 pesticides but just five detected. and processing factor should be caculated with different processing step or parts. The results are too few data to discuss.
Author Response
Reviewer 2
It is very interesting to study the effect of pesticide residue in artichoke during industrial processing with queQuEChERS method.
1)It would be better if use field sample with pesticide spraying, not just could determine 220 pesticides but just five detected.
Answer: The analysis were carried out on real field, during the harvesting time, in accordance with the processing industry. To perform such a study with unauthorized pesticides on artichoke, the entire field should be discarded as special waste and the field should have been sanitized after eliminating the artichokes.
2) and processing factor should be calculated with different processing step or parts.
Answer: the processing factor was calculated only for the outer bracts considering that they represent the 95% of the waste from the industrial processing. For residue value < LOQ, no PF was calculated. Comments were added in the discussion and a column in table 1 (PF).
3)The results are too few data to discuss.
Answer: the results from the field are few, but this is a good information. However, some pesticides were not used in this season on artichoke and probably belong from previous treatments.
Reviewer 3 Report
In this study the authors analyze 220 different compounds and mention that of the 220, 5 were found on the samples they collected. I would like to have more information about agricultural chemicals applied to the field during growing. Were there ones that were applied that were not detected?
Figure 1 should have bigger font for the axes as it is completely unreadable.
Author Response
Reviewer 3
In this study the authors analyze 220 different compounds and mention that of the 220, 5 were found on the samples they collected. I would like to have more information about agricultural chemicals applied to the field during growing. Were there ones that were applied that were not detected?
Answer: The field evaluated in this study, was subjected with treatments with pendimethalin and azoxystrobin. Propyzamid, tebuconazole, and pyraclostrobin were not applied during the current producing season. Therefore, the presence of residues was related to treatments on artichoke or on other crops cultivated in the same field in the previous year. This topic was addressed along the discussion section.
Figure 1 should have bigger font for the axes as it is completely unreadable.
Answer: Figure 1 was extrapolated from the instrument without further processing by software.
Round 2
Reviewer 2 Report
It is very important to detect the pesticide residues in IV range artichoke (Cynara cardunculus L) and find the processing factor for residues. it was greatly impoved after the major revision, which could be suitable to publication now.